# Controlling Drain Side Tunneling Barrier Width in Electrically Doped PNPN Tunnel FET

**DOI:** 10.3390/mi14020301

**Published:** 2023-01-24

**Authors:** Chan Shan, Lan Yang, Ying Liu, Zi-Meng Liu, Han Zheng

**Affiliations:** 1College of Ocean Information Engineering, Jimei University, Xiamen 361021, China; 2College of Software, Quanzhou University of Information Engineering, Quanzhou 362000, China; 3Department of Software Technology, Xiamen Institute of Software Technology, Xiamen 361021, China

**Keywords:** electrically doping, PNPN tunnel FET, ambipolar current, tunneling barrier width, TCAD simulation

## Abstract

In this paper, we propose and investigate an electrically doped (ED) PNPN tunnel field effect transistor (FET), in which the drain side tunneling barrier width is effectively controlled to obtain a suppressed ambipolar current. We present that the proposed PNPN tunnel FETs can be realized without chemically doped junctions by applying the polarity bias concept to a doped N^+^/P^−^ starting structure. Using numerical device simulations, we demonstrate how the tunneling barrier width on the drain side can be influenced by several design parameters, such as the gap length between the channel and the drain (L_gap_), the working function of the polarity gate, and the dielectric material of the spacer. The simulation results show that an ED PNPN tunneling FET with an ED drain, which has been explored for the first time, exhibits a low ambipolar current of 5.87 × 10^−16^ A/µm at a gap length of 20 nm. The ambipolar current is reduced by six orders of magnitude compared to that which occurs with a conventional ED PNPN tunnel FET with a uniformly doped drain, while the average subthreshold slope and the ON state and OFF state currents remained nearly identical.

## 1. Introduction

Tunnel field effect transistors (TFETs) are promising low power devices that can replace metal oxide semiconductor field effect transistors (MOSFETs) in ultra low power circuits applications due to the subthreshold swing (SS) of less than 60 mV/decade at room temperature and a lower OFF state current [1,2,3,4,5,6]. However, due to the inherent band-to-band tunneling and ambipolar mechanism, TFETs also have their own drawbacks in terms of a low drive current and a high ambipolar current. Ambipolarity refers to the device conducting both positive and negative gate voltages. In complementary circuit applications, the TFET’s ambipolar conduction limits its usefulness [7,8,9]. PNPN tunnel FETs, which have a narrow N^+^ pocket between the source and the channel, have been intensively investigated to improve their electrical performances [10,11,12,13,14]. Compared to the conventional TFETs, PNPN tunnel FETs have shown their advantages such as a higher drive current, an enhanced SS, and an improved device reliability. However, realizing this narrow N+ doped pocket is a technological challenge [15,16,17,18]. Based on the polarity bias concept, we have recently reported a theoretical device structure of an in-built N^+^ pocket electrically doped (ED) tunnel FET, which provides a possible technique to overcome the difficulties in realizing a highly doped narrow pocket [19].

By using the polarity bias concept [20,21,22], a conventional in-built N^+^ pocket electrically doped tunnel FET is presented, as shown in Figure 1b. With a starting NPN structure (shown in Figure 1a), a polarity gate electrode is set on the source side of the device to achieve the PNPN TFET structure. Therefore, the narrow N^+^ pocket is built in the device and without the need for any additional doping processes, which simplifies the manufacturing complexity. Furthermore, 2D Technology Computer Aided Design (TCAD) simulations have demonstrated that the introduction of the in-built N^+^ pocket yields a higher drive current and a steeper average SS when they are compared to those of a conventional ED-TFET. However, the ambipolar characteristics of the in-built N^+^ pocket ED-TFET have not been discussed yet.

In this paper, in order to perform a further analysis on the ambipolar characteristics and suppress the ambipolar current by controlling the tunneling barrier width at channel–drain junction, we propose and investigate an in-built N^+^ pocket ED-TFET with an electrically doped drain. To realize the proposed device, first, a PN junction structure is depicted, as shown in Figure 1c. The proposed device is composed of three sets of gate electrodes: a control gate (CG) and two polarity gates (PG). Then, one polarity gate electrode (marked as PG1) is embedded on the source side of the device to convert part of the N^+^ doped source into a “P^+^” region. Another polarity gate electrode (marked as PG2) is embedded on the drain side to convert part of the p-doped region into an “N^+^” drain [21]. Therefore, an in-built N^+^ pocket ED-TFET with an ED drain is realized, as shown in Figure 1d. To create a P^+^ source region, the PG1 terminal is biased at −0.7 V to increase the hole concentration to 1 × 10^19^ cm^−3^ in the TCAD simulations. Meanwhile, to create an N^+^ drain region, the PG2 terminal is biased at 0.7 V to increase the electron concentration to 5 × 10^18^ cm^−3^. The source and drain contacts are composed of nickel silicide (NiSi) with a Schottky barrier height of 0.45 eV. Band bending at the source/drain contact indicates a low PG bias, thereby making the holes become the majority carriers, and a high PG bias makes the electrons become the majority carriers, as well [23]. Thus, to create a P^+^ source region with a hole concentration that is similar to that of the reference device, PG1 is biased sufficiently negative. Furthermore, PG2 is sufficiently positively biased for an N^+^ drain region with an electron concentration that is equivalent to that of the reference device.

We can see from Figure 1b,d that the main difference between these two structures is the formation of the drain region. In the conventional in-built N^+^ pocket ED-TFET, the N+ drain region is uniformly doped by ion implantation [19]. However, in the proposed device with an ED drain, the drain region is electrically doped based on the polarity bias concept, and it introduces a new design parameter L_gap_, which is the length of the gap between the channel and the drain. Table 1 is presented to compare the proposed device with the conventional in-built N+ pocket ED-TFET and the conventional ED-TFET structures. The quotation marks “P^+^” and “N^+^” in the doping profile of the final structure indicate that this region is formed by electrical doping using the polarity bias concept, rather than ion implantation. The simulation results show that the tunneling barrier width at the channel–drain junction can be controlled by choosing an appropriate L_gap_ for the proposed device. Thus, the ambipolar current of the device can be effectively suppressed by introducing a gap on the drain side by using an electrical doping technique based on the concept of polarity bias.

## 2. Simulation Parameters and Approach

The device parameters used in our simulations are shown in Table 2. In order to study the effect of the gap length on the tunneling barrier width and ambipolar current, L_gap_ varies from 5 nm to 25 nm. Making the layer underneath the polarity gate intrinsic, the working functions of PG1 and PG2 in the proposed device with an ED drain were chosen to be 4.33 eV and 4.5 eV, which refer to the Ti and Mo materials, respectively [24]. An HfO_2_ gate dielectric with a physical thickness of approximately 4.5 nm and an equivalent oxide thickness (EOT) of 0.8 nm was used in the simulations. The source terminals 0were set to be grounded (V_S_ = 0) in all of the simulations. Hence, we have regarded V_CG_ = V_CGS_ and V_PG_ = V_PGS_ in all of the devices.

All of the simulations were performed using the Silvaco Atlas device simulation tool, version 5.19.20.R [25]. The non-local band-to-band tunneling (BTBT) model was used to take into account the tunneling along the lateral direction. We used the Lombardi mobility model in our simulation in order to include the mobility degradation effect owing to the electric field. The Shockley–Read–Hall (SRH) recombination model and Fermi Dirac statistics were also used. The band-gap narrowing (BGN) model was used for the high doping concentration in the devices. Quantum mechanical effects were not considered when the *T*_Si_ exceeded 7 nm [5,26]. Thus, quantum mechanical effects are not included in our simulations since *T*_Si_ was designed to be 10 nm. Based on [4], we validated our simulation model using a nonlocal band-to-band tunneling (BTBT) model. Non-local BTBT computed the tunneling probability along the lateral direction of the device by analyzing the energy band diagrams. In the simulations, non-local BTBT was performed using a fine mesh across the region where tunneling occurs. A careful application of the Wentzel–Kramer–Brillouin method was used by Atlas for approximating the evanescent wavevector. There was no consideration of the gate leakage in these simulations, which can be expected to limit the OFF current. The bias condition is that the voltage of PG1 is fixed at −0.7 V, the voltage of PG2 is fixed at 0.7 V, and the drain voltage is fixed at 1 V. V_CG_ increased from −1 V to 1 V.

## 3. Results and Discussion

Figure 2 shows the transfer characteristics of the proposed in-built N^+^ pocket ED-TFET with an ED drain and a conventional device with uniform doped drain. The effect of changing the gap length L_gap_ on the ambipolar conduction is also shown in Figure 2. Accordingly, this gap corresponds to the channel-to-drain depletion region in the conventional TFETs. The drain side depletion region should be wide enough to minimize the ambipolar currents. We can see from Figure 2 that for the proposed ED drain device, the ambipolar current is significantly suppressed for the control gate voltage that is as large as −1.0 V, and the gap length reaches 10 nm. It can be seen that the bipolar current is effectively controlled whenever the gap length increases, as the gap length determines the tunneling barrier width in the drain region. A larger L_gap_ is equivalent to a wider depletion width, and thus, it increases the tunneling barrier width at the channel–drain junction. Therefore, by choosing an appropriate gap length, the ambipolar current can be effectively suppressed. As shown in Figure 3, we present the energy band profiles of the proposed device below the Si–oxide interface at 1 nm and 5 nm with different gap lengths to explain the reduction in the ambipolar current with gap length. According to Figure 3, with a gap length of 20 nm, the tunnel barrier width is larger due to a wider depletion region between the channel and drain, resulting in a suppressed ambipolar conduction. The electron concentration contour plot of the proposed in-built N^+^ pocket ED-TFET with an ED drain for L_gap_ values of 5 nm and 20 nm is shown in Figure 4 at V_CG_ = −1 V. We observe that the electron concentration varies drastically at the gaps. When the gap length is 5 nm, the electron concentration of the drain region near the channel side is higher, which is consistent with the performance of the energy band diagram in Figure 3. For different PG2 working functions, the transfer characteristics of the proposed in-built N^+^ pocket ED-TFET with an ED drain is shown in Figure 5. The gap length is fixed at 5 nm. The working functions of PG2 are set to 4.33 eV, 4.5 eV, and 4.74 eV, referring to Ti, Mo, and Sb, respectively [24]. In Figure 5, we observe that the ambipolar current continuously decreases, while the OFF state current remains nearly constant with an increasing Φ_PG2_. The ambipolar current (I_AMB_) is extracted at V_CG_ = −0.8 V, and the OFF state current (I_OFF_) is extracted at V_CG_ = 0.0 V. To understand this decrease in the ambipolar current, the energy band profile of the proposed in-built N^+^ pocket ED-TFET with an ED drain with a different Φ_PG2_ is shown in Figure 6. As Φ_PG2_ increases, the induced electron concentration in the drain region decreases due to the polarity bias concept. Thus, a wider tunnel barrier width is seen in Figure 6 for Φ_PG2_ of 4.74 eV, which results in a suppressed ambipolar current. Figure 7 shows transfer characteristic curves of the proposed in-built N^+^ pocket ED-TFET with an ED drain for different spacer materials at the drain side. As it can be seen from Figure 7, the spacer material has a small effect on the ambipolar conduction of the device, and it does not affect the electrical characteristics in the OFF state and ON state at all. The effect on the ambipolar conduction is that the higher the dielectric constant of the spacer is, the lower the ambipolar current is. To further explain the reason, we present the energy band diagrams for both Air and HfO_2_ spacer material devices, as shown in Figure 8. The widths of the tunneling barrier corresponding to the two spacer materials are shown in the enlarged image in Figure 8, which are 6 nm for the Air spacer and 7 nm for the HfO_2_ spacer. Additionally, this explains the results mentioned above. Therefore, this study indicates that the in-built N^+^ pocket ED-TFET with an ED drain offers a new method of suppressing the ambipolar currents in TFETs, as well as a low thermal budget process since the drain region does not need to be chemically doped.

## 4. Conclusions

In this paper, with the help of 2D TCAD simulations, we present an approach that can be used to realize an in-built N^+^ pocket ED-TFET with an electrically doped drain. Based on polarity bias concept, we have shown that the tunneling barrier width at the drain side can be effectively controlled by several design parameters. The simulation results reveal that a PNPN tunneling FET with an ED drain, which has been presented for the first time, exhibits a low ambipolar current of 5.87 × 10^−16^ A/µm over a gap length of 20 nm. As compared to the conventional ED PNPN tunneling FETs with uniformly doped drains, the ambipolar current is reduced by six orders of magnitude, while the average subthreshold slope, ON state current, and OFF state current remain nearly the same. As a result, the ambipolar current of the proposed device can be effectively suppressed by choosing appropriate design parameters.

## Figures and Tables

**Figure 1 micromachines-14-00301-f001:**
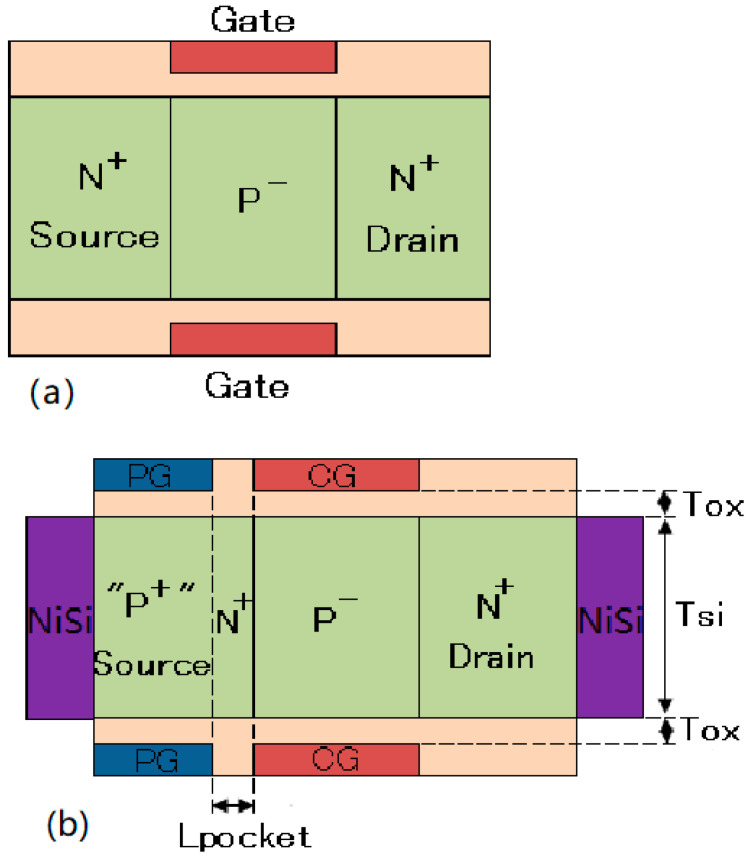
Cross-sectional views of (**a**) initial NPN structure used to realize conventional in-built N^+^ pocket ED-TFET, (**b**) conventional in-built N^+^ pocket ED-TFET, (**c**) initial NP structure used to realize in-built N^+^ pocket ED-TFET with an ED drain, and (**d**) in-built N^+^ pocket ED-TFET with an ED drain.

**Figure 2 micromachines-14-00301-f002:**
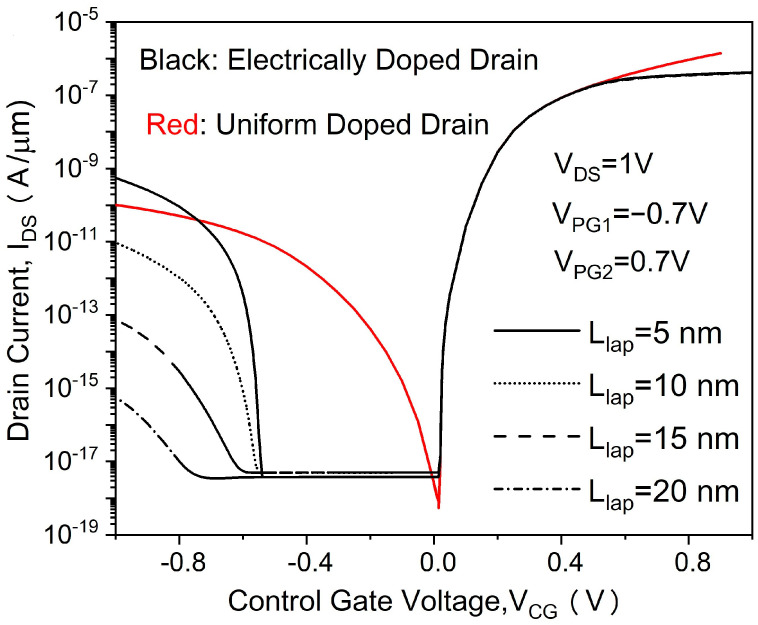
Transfer characteristics of proposed in-built N^+^ pocket ED-TFET with an ED drain and conventional in-built N^+^ pocket ED-TFET with uniformly doped drain.

**Figure 3 micromachines-14-00301-f003:**
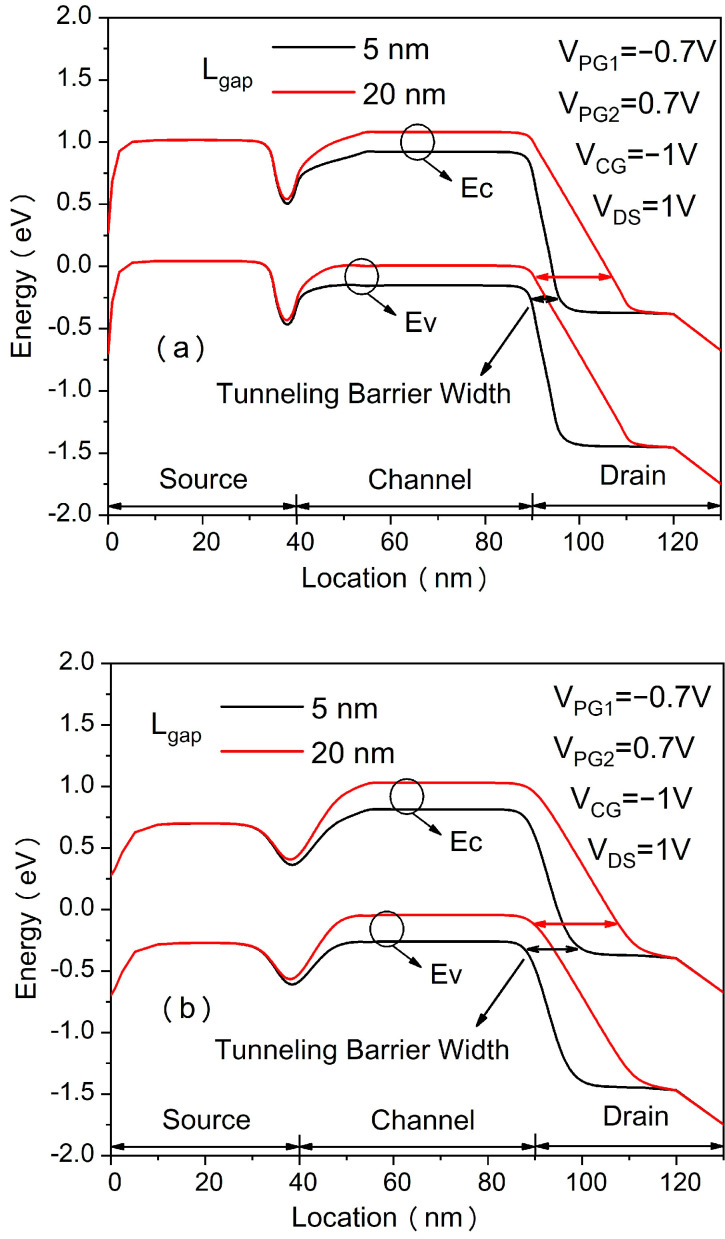
Energy band diagrams at (**a**) 1 nm and (**b**) 5 nm below the Si–oxide interface of the proposed ED-TFET with an ED drain at OFF state with Lgap of 5 nm (black) and 20 nm (red).

**Figure 4 micromachines-14-00301-f004:**
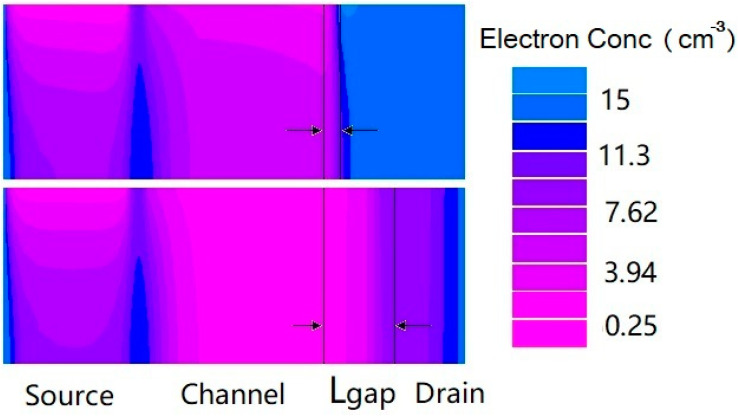
Electron concentration contour of proposed in-built N^+^ pocket ED-TFET with an ED drain for L_gap_ of 5 nm (**up**) and 20 nm (**down**) at OFF state.

**Figure 5 micromachines-14-00301-f005:**
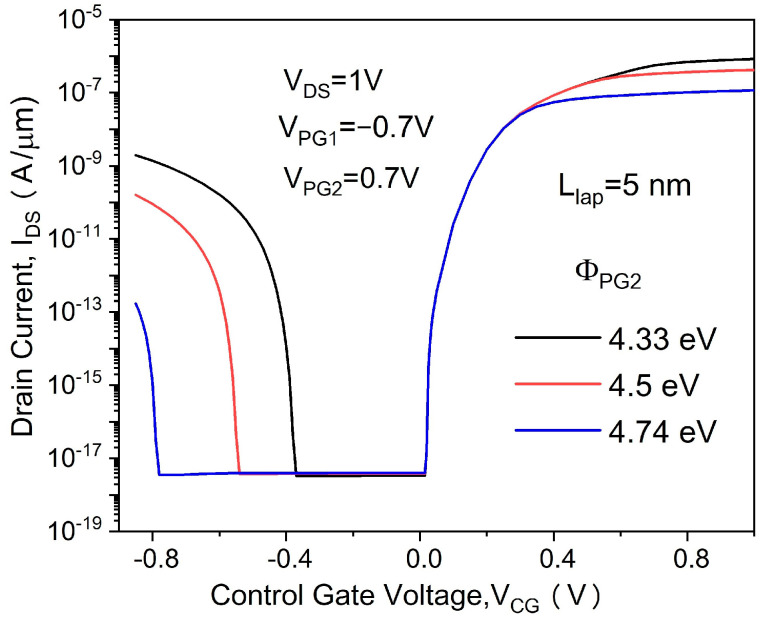
Transfer characteristics of proposed in-built N^+^ pocket ED-TFET with an ED drain for different PG2 working functions.

**Figure 6 micromachines-14-00301-f006:**
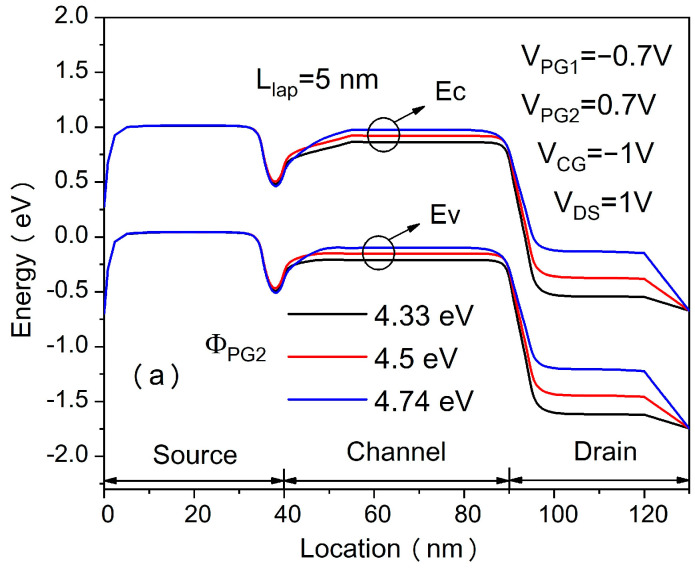
Energy band diagrams at (a) 1 nm and (**b**) 5 nm below the Si–oxide interface of the proposed ED-TFET with an ED drain at OFF state with ΦPG2 of 4.33eV (black), 4.5eV (red) and 4.74eV (blue).

**Figure 7 micromachines-14-00301-f007:**
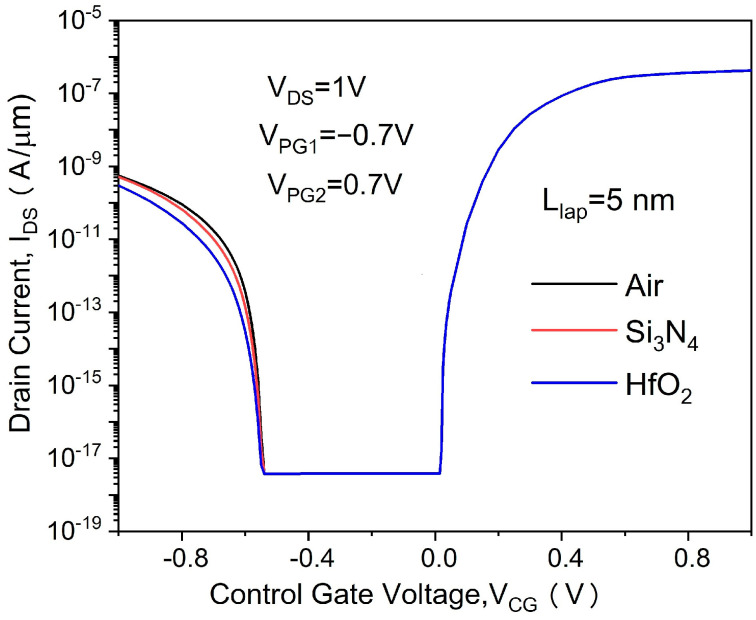
Transfer characteristics of proposed in-built N^+^ pocket ED-TFET with an ED drain for different spacer materials at the drain side.

**Figure 8 micromachines-14-00301-f008:**
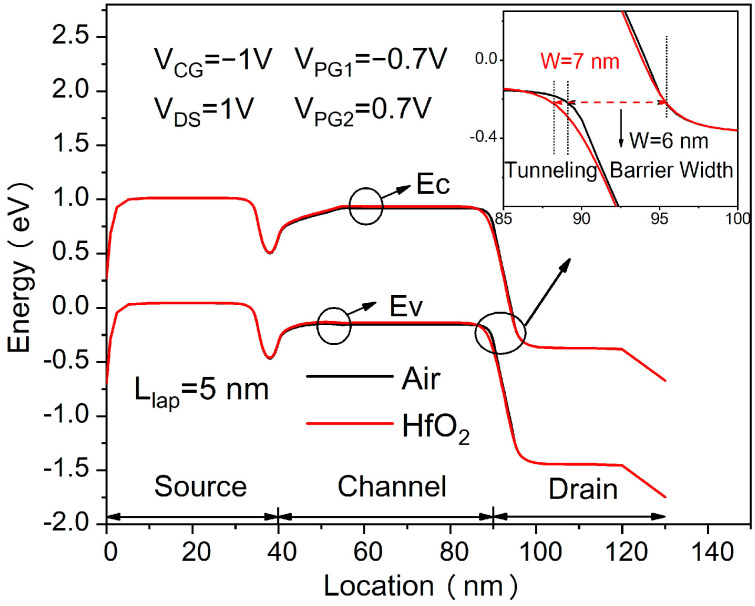
Energy band diagrams of the proposed ED-TFET with an ED drain with Air and HfO_2_ spacer at the drain side.

**Table 1 micromachines-14-00301-t001:** Comparison of device design.

Device Design	Conventional ED-TFET	Conventional In-Built N^+^ Pocket ED-TFET with Uniform Doped Drain [19]	Proposed In-Built N^+^ Pocket ED-TFET with an ED Drain
Doping Profile of Beginning Structure (From Source to Drain)	N^+^-N^+^-N^+^	N^+^-P^−^-N^+^	N^+^-P^−^
Doping Profile of Final Stucture (From Source to Drain)	“P^+^”-N^+^-N^+^-N^+^	“P^+^”-N^+^-P^−^ -N^+^	“P^+^”-N^+^-P^−^-“N^+^”
Number of Polarity Gate	1	1	2
Location of Polarity Gate	Source Region	Source Region	Source and Drain Region

**Table 2 micromachines-14-00301-t002:** Parameters used for device simulation.

Parameter	Conventional In-Built N^+^ Pocket ED-TFET [19]	In-Built N^+^ Pocket ED-TFET with an ED Drain
Effective Gate Oxide Thickness (*EOT*)	0.8 nm	0.8 nm
Silicon Film Thickness (*T*_Si_)	10 nm	10 nm
Control Gate Length	50 nm	50 nm
Length of pocket (*L*_pocket_)	5 nm	5 nm
Length of gap (*L*_gap_)	-	5~25 nm
Source Doping	4 × 10^19^ cm^−3^ (N^+^)	4 × 10^19^ cm^−3^ (N^+^)
Channel Doping	1 × 10^17^ cm^−3^ (P^−^	1 × 10^17^ cm^−3^ (P^−^)
Drain Doping	1 × 10^17^ cm^−3^ (P^−^)	5 × 10^18^ cm^−3^ (N^+^)
CG Work function	4.74 eV	4.74 eV
PG1 Work function	4.33 eV	4.33 eV
PG2 Work function	-	4.5 eV

## Data Availability

The data presented in this study are available on request from the corresponding author. The data are not publicly available.

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
