# Peer review of "Controlling Drain Side Tunneling Barrier Width in Electrically Doped PNPN Tunnel FET"

_micromachines, 2023, doi:10.3390/mi14020301_

Round 1

Reviewer 1 Report

The authors have introduced a new design parameter namely Lgap for controlling the ambipolar current of a TFET which is an improvement to their previously published paper in micromachines. The introduced concept is new, but the following comments should be addressed to make the paper ready for publication and enhance its quality.

1)     The concept seems new, but authors should prepare a table in which they compared the design with their previous design (without PG2) and conventional TFET. This would further highlight the novelty of this paper compared to the previous paper(s). Also, authors should clearly state their previous work (see comment 4) and bring up their new contribution in this paper.

2)     In the abstract authors say, “Using calibrated TCAD simulations, we demonstrate how ….”, but there is no place in the manuscript where they showed the calibration of the TCAD simulation or numerical method with an experiment or other simulation, authors should open a new section or add it in the appendix. It should be explained somewhere in the manuscript.

3)     Also, please open a section and explain more about the simulation done with TCAD, the software used, meshing, setup, and configuration or biasing condition.

4)     I think the abstract and conclusion do not reflect the novelties of this work properly and efficiently. It is advised that authors talk about performance enhancement or another figure of merit (like transconductance, SS or …) and generally an improvement in this paper to better highlight the novelties in this work. The abstract and conclusion should be improved.

5)     Why did the PG1 create the P+ region, please add a reference or explain in the text.

6)     It should be clarified in the manuscripts why the same but different polarity voltage of 0.7 and -0.7 at both ends of silicon has created different doping concentrations, 1*10^19 and 5*10^18. Probably different work functions in place of junctions? More explanations are required in the part of the paper (lines 50-56).

7)     For this sentence “In conventional in-built N+ pocket 64 ED-TFET, the N+ drain region is uniformly doped by ion implantation” a reference is needed, probably here is where the author’s other paper with the same topic (discussing the same structure without drain part, PG2), the paper is

-        Li, J.; Liu, Y.; Wei, S.-f.; Shan, C. In-Built N+ Pocket Electrically Doped Tunnel FET With Improved DC and Analog/RF Performance. Micromachines 202011, 960. https://doi.org/10.3390/mi11110960

8)     previous knowledge is required to better lead the reader to understand “Thus, the ambipolar current of the device can be effectively sup- 70 pressed by introducing a gap on the drain side by using electrically doping technique 71 based on the concept of polarity bias”. Please talk about it in previous parts of the introduction why surpassing ambipolar current is required in this design.

9)     Which figure authors mean? in “We can see from the figure (?) that, for the proposed ED drain device, the 98 ambipolar currents are significantly suppressed for the control gate voltage as large as -1.0 99 V and the gap length reaches 10 nm”.

10)  Authors mentioned, “Therefore, based on the amount of ambipolar current 102 that is acceptable for a given application, the gap length can be selected.” This might not be a good selling point for this length.

11)  Please specify the figure everywhere in the manuscript like this one which is ambiguous: “As can be seen from the figure, the spacer material has 121 a small effect on the ambipolar conduction of the device and does not affect the electrical 122 characteristics in the OFF-state and ON-state at all”

12)  The unit for current in figure 6 and figure 4 is absolute or it is normalized with the 3D dimension of the width of the device. A or A/um?

13)  Please add the simulation result for instance the current contours in 2D or the doping concentration somewhere in the paper to show the gap effects in better presentation.

14)  What do the authors mean in the conclusion by saying “but also has a low thermal budget because of 155 of the electrically doped drain”? Thermal budget? Is that a figure of merit for device comparison?

Author Response

Dear editor,

   We are truly grateful to yours and other reviewers’ critical comments and thoughtful suggestions. Based on these comments and suggestions, we have made careful modifications on the original manuscript. We hope the new manuscript will meet your magazine’s standard. Below you will find our point-by-point responses to the reviewers’ comments/ questions:

Reviewer 1:

The authors have introduced a new design parameter namely Lgap for controlling the ambipolar current of a TFET which is an improvement to their previously published paper in micromachines. The introduced concept is new, but the following comments should be addressed to make the paper ready for publication and enhance its quality.

1)     The concept seems new, but authors should prepare a table in which they compared the design with their previous design (without PG2) and conventional TFET. This would further highlight the novelty of this paper compared to the previous paper(s). Also, authors should clearly state their previous work (see comment 4) and bring up their new contribution in this paper.

  1. Thank you for your suggestion. A table has been added based on your advice.

Table 1 is presented to compare the proposed device with conventional in-built N+ pocket ED-TFET and conventional ED-TFET structures. The quotation marks "P+" and "N+" in the doping profile of the final structure indicate that this region is formed by electrically doping using the polarity bias concept, rather than ion implantation.

Table 1. Comparison of device design.

Device Design

Conventional ED-TFET

Conventional in-built N+ pocket ED-TFET [19]

In-built N+ pocket ED-TFET with an ED drain

Doping Profile of Beginning Structure (From Source to Drain)

N+- N+- N+

N+- P- - N+

N+- P-

Doping Profile of Final Stucture (From Source to Drain)

“P+”- N+- N+- N+

“P+”- N+- P- - N+

“P+”- N+- P- -“N+

Number of Polarity Gate

1

1

2

Location of Polarity Gate

Source Region

Source Region

Source and Drain Region

2)     In the abstract authors say, “Using calibrated TCAD simulations, we demonstrate how ….”, but there is no place in the manuscript where they showed the calibration of the TCAD simulation or numerical method with an experiment or other simulation, authors should open a new section or add it in the appendix. It should be explained somewhere in the manuscript.

  1. Thank you for your suggestion. The manuscript has been modified based on your advice.

Based on Ref. [4], we validate our simulation model using a nonlocal band-to-band tunneling (BTBT) model. Nonlocal BTBT computes the tunneling probability along the lateral direction of the device by analyzing energy-band diagrams. In the simulations, nonlocal BTBT is performed using a fine mesh across the region where tunneling occurs. A careful application of Wentzel–Kramer–Brillouin method is used by Atlas for approximating the evanescent wavevector. There was no consideration of gate leakage in these simulations, which can be expected to limit the OFF-current. The bias condition is that the voltage of PG1 is fixed at -0.7V, the voltage of PG2 is fixed at 0.7V and the drain voltage is fixed at 1V. VCG increases from -1V to 1V.

3)     Also, please open a section and explain more about the simulation done with TCAD, the software used, meshing, setup, and configuration or biasing condition.

  1. Thank you for your suggestion. The manuscript has been modified based on your advice.

All the simulations are done using Silvaco Atlas device simulation tool, version 5.19.20.R [25].

Based on Ref. [4], we validated our simulation model using a nonlocal band-to-band tunneling (BTBT) model. Nonlocal BTBT computes the tunneling probability along the lateral direction of the device by analyzing energy-band diagrams. In the simulations, nonlocal BTBT is performed using a fine mesh across the region where tunneling occurs. A careful application of Wentzel–Kramer–Brillouin method is used by Atlas for approximating the evanescent wavevector. There was no consideration of gate leakage in these simulations, which can be expected to limit the OFF-current. The bias condition is that the voltage of PG1 is fixed at -0.7V, the voltage of PG2 is fixed at 0.7V and the drain voltage is fixed at 1V. VCG increases from -1V to 1V.

4)     I think the abstract and conclusion do not reflect the novelties of this work properly and efficiently. It is advised that authors talk about performance enhancement or another figure of merit (like transconductance, SS or …) and generally an improvement in this paper to better highlight the novelties in this work. The abstract and conclusion should be improved.

  1. Thank you for your suggestion. The abstract and conclusion has been modified based on your advice.

Abstract: In this paper, we propose and investigate an electrically doped (ED) PNPN tunnel field effect transistor (FET), in which the drain side tunneling barrier width is effectively controlled to obtain suppressed ambipolar current. We present that proposed PNPN tunnel FETs can be realized without chemically doped junctions by utilizing polarity bias concept on a doped N+/P- starting structure. Using numerical device simulations, we demonstrate how the tunneling barrier width on the drain side can be influenced by several design parameters, such as the gap length between channel and drain (Lgap), the work-function of polarity gate, and dielectric material of the spacer. The simulation results show that an ED PNPN tunneling FET with an ED drain, which is explored for the first time, exhibits a low ambipolar current of 5.87◊10-16 A/µm at a gap length of 20 nm. The ambipolar current is reduced by 6 orders of magnitude compared to a conventional ED PNPN tunnel FET with a uniformly doped drain, while the average subthreshold slope, on-state and off-state currents remain nearly identical.

Conclusions

In this paper, with the help of 2D TCAD simulations, we present an approach to realize an in-built N+ pocket ED-TFET with an electrically doped drain. Based on polarity bias concept, we have shown that the tunneling barrier width at the drain side can be effectively controlled by several design parameters. Simulation results reveal that a PNPN tunneling FET with an ED drain, presented for the first time, exhibits a low ambipolar current of 5.87◊10-16 A/µm over a gap length of 20 nm. As compared to conventional ED PNPN tunneling FETs with uniformly doped drains, the ambipolar current is reduced by six orders of magnitude, while the average subthreshold slope, on-state current, and off-state current remain nearly the same. As a result, the ambipolar current of the proposed device can be effectively suppressed by choosing appropriate design parameters.

5)     Why did the PG1 create the P+ region, please add a reference or explain in the text.

  1. Thank you for your suggestion. The manuscript has been modified based on your advice.

Then one polarity gate electrode (marked as PG1) is embedded on the source side of the device to convert part of the N+ doped source into a “P+” region. Another polarity gate electrode (marked as PG2) is embedded on the drain side to convert part of the p- doped region into an “N+” drain [21]. Band bending at the Source/Drain contacts indicates low PG bias making holes become the majority carriers, and high PG bias making electrons become the majority carriers, as well [23]. Thus, to create a P+ source region with hole concentration similar to a reference device, PG1 is biased sufficiently negative. Furthermore, PG2 is sufficient positive biased for an N+ drain region with an electron concentration equivalent to the reference device.

In Ref. [21], “Furthermore, for creating a n+ drain region having electron concentration similar to a reference device, sufficient positive bias is applied at PG-1 (50 nm). Similarly, for a p+ source region having hole concentration equivalent to the reference device, sufficient negative bias is applied at PG-2 (50 nm).”

  1. Lahgere, A.; Sahu, C.; Singh, J. PVT-aware design of dopingless dynamically configurable tunnel FET. IEEE Trans. Electron Dev. 2015, 62, 2404–2409. DOI: 10.1109/TED.2015.2446615.
  2. De Marchi, M.; Sacchetto, D.; Frache, S.; Zhang, J.; Gaillardon, P.-E.; Leblebici, Y.; De Micheli, G. Polarity control in double-gate, gate-all-around vertically stacked silicon nanowire FETs. in Proceedings of IEEE Electron Devices Meeting, San Francisco, CA, USA, 10 Dec. 2012. DOI: 10.1109/IEDM.2012.6479004.

6)     It should be clarified in the manuscripts why the same but different polarity voltage of 0.7 and -0.7 at both ends of silicon has created different doping concentrations, 1*10^19 and 5*10^18. Probably different work functions in place of junctions? More explanations are required in the part of the paper (lines 50-56).

  1. Thank you for your suggestion. The manuscript has been modified based on your advice.

Band bending at the Source/Drain contacts indicates low PG bias making holes become the majority carriers, and high PG bias making electrons become the majority carriers, as well [23]. Thus, to create a P+ source region with hole concentration similar to a reference device, PG1 is biased sufficiently negative. Furthermore, PG2 is sufficient positive biased for an N+ drain region with an electron concentration equivalent to the reference device.

  1. De Marchi, M.; Sacchetto, D.; Frache, S.; Zhang, J.; Gaillardon, P.-E.; Leblebici, Y.; De Micheli, G. Polarity control in double-gate, gate-all-around vertically stacked silicon nanowire FETs. in Proceedingds of IEEE Electron Devices Meeting (IEDM), San Francisco, CA, USA, 10 Dec. 2012. DOI: 10.1109/IEDM.2012.6479004.

7)     For this sentence “In conventional in-built N+ pocket 64 ED-TFET, the N+ drain region is uniformly doped by ion implantation” a reference is needed, probably here is where the author’s other paper with the same topic (discussing the same structure without drain part, PG2), the paper is

-        Li, J.; Liu, Y.; Wei, S.-f.; Shan, C. In-Built N+ Pocket Electrically Doped Tunnel FET With Improved DC and Analog/RF Performance. Micromachines 202011, 960. https://doi.org/10.3390/mi11110960

  1. Thank you for your suggestion. The reference has been added based on your advice.

In conventional in-built N+ pocket ED-TFET, the N+ drain region is uniformly doped by ion implantation [19].

  1. Li, J.; Liu, Y.; Wei, S. F.; Shan, C. In-Built N+ Pocket Electrically Doped Tunnel FET With Improved DC and Analog/RF Performance. Micromachines, 2020, 11, 960. DOI: 10.3390/mi11110960.

8)     previous knowledge is required to better lead the reader to understand “Thus, the ambipolar current of the device can be effectively sup- 70 pressed by introducing a gap on the drain side by using electrically doping technique 71 based on the concept of polarity bias”. Please talk about it in previous parts of the introduction why surpassing ambipolar current is required in this design.

  1. Thank you for your suggestion. The manuscript has been modified based on your advice.

However, due to the inherent band-to-band tunneling and ambipolar mechanism, TFETs also have their own drawbacks, in terms of low drive current and high ambipolar current. Ambipolarity refers to the device conducting for both positive and negative gate voltages. In complementary circuit applications, the TFET's ambipolar conduction limits its usefulness [7-9].

9)     Which figure authors mean? in “We can see from the figure (?) that, for the proposed ED drain device, the 98 ambipolar currents are significantly suppressed for the control gate voltage as large as -1.0 99 V and the gap length reaches 10 nm”.

  1. We are so sorry for the mistake. The manuscript has been modified based on your advice.

We can see from Fig. 2 that, for the proposed ED drain device, the ambipolar current is significantly suppressed for the control gate voltage as large as -1.0 V and the gap length reaches 10 nm.

10)  Authors mentioned, “Therefore, based on the amount of ambipolar current 102 that is acceptable for a given application, the gap length can be selected.” This might not be a good selling point for this length.

  1. Thank you for your suggestion. The manuscript has been modified based on your advice.

A larger Lgap is equivalent to a wider depletion width and thus increases the tunneling barrier width at the channel–drain junction. Therefore, by choosing an appropriate gap length, the ambipolar current can be effectively suppressed.

11)  Please specify the figure everywhere in the manuscript like this one which is ambiguous: “As can be seen from the figure, the spacer material has 121 a small effect on the ambipolar conduction of the device and does not affect the electrical 122 characteristics in the OFF-state and ON-state at all”

  1. We are so sorry for the mistake. The manuscript has been modified based on your advice.

We can see from Fig. 2 that, for the proposed ED drain device, the ambipolar current is significantly suppressed for the control gate voltage as large as -1.0 V and the gap length reaches 10 nm.

As can be seen from Fig. 7, the spacer material has a small effect on the ambipolar conduction of the device and does not affect the electrical characteristics in the OFF-state and ON-state at all.

12)  The unit for current in figure 6 and figure 4 is absolute or it is normalized with the 3D dimension of the width of the device. A or A/um?

  1. We are so sorry for the mistake. The manuscript has been modified based on your advice.

The unit for current in the figure is modified to A/um.

Figure 2. Transfer characteristics of proposed in-built N+ pocket ED-TFET with an ED drain and conventional in-built N+ pocket ED-TFET with a uniformly doped drain.

Figure 5. Transfer characteristics of proposed in-built N+ pocket ED-TFET with an ED drain for different PG2 work-functions.

Figure 7. Transfer characteristics of proposed in-built N+ pocket ED-TFET with an ED drain for different spacer materials at the drain side.

13)  Please add the simulation result for instance the current contours in 2D or the doping concentration somewhere in the paper to show the gap effects in better presentation.

  1. Thank you for your suggestion. The manuscript has been modified based on your advice.

The electron concentration contour plot of proposed in-built N+ pocket ED-TFET with an ED drain for Lgap of 5nm and 20nm is shown in Fig.4 at VCG = −1 V. We observe that the electron concentration varies drastically at the gaps. When the gap length is 5 nm, the electron concentration of the drain region near the channel side is higher, which is consistent with the performance of the energy band diagram in Fig. 3.

Figure 4. Electron concentration contour of proposed in-built N+ pocket ED-TFET with an ED drain for Lgap of 5nm (up) and 20nm (down) at OFF-state.

14)  What do the authors mean in the conclusion by saying “but also has a low thermal budget because of 155 of the electrically doped drain”? Thermal budget? Is that a figure of merit for device comparison?

  1. We are so sorry for the mistake. The sentence of “but also has a low thermal budget because of 155 of the electrically doped drain” in the conclusion has been removed since we have not used the software to put some numerical data about the thermal behavior of the modeled FET.

As a result, the ambipolar current of the proposed device can be effectively suppressed by choosing appropriate design parameters.

Reviewer 2 Report

Abstract:

-          The concept of “electrical doped” from the sentence “In this paper, we propose and investigate an electrically doped (ED) PNPN tunnel FET…” must be clearly explained. Usually the term “doped” refers to an amount of another chemical element/compound added in order to obtain a certain effect.

-          All acronyms must be explained when firstly introduced, e.g. FET (field-effect transistor (FET))

-          The concept/method of “calibrated TCAD simulations” must be explained as not usually used

1. Introduction

The authors claim that “Based on polarity bias concept, we have recently reported an in-built N+ pocket electrically doped (ED) tunnel FET, which 31 provides a possible technique to overcome the difficulties in realizing a highly doped 32 narrow pocket [16].” In this sense, they are kindly asked to explain if “in-built N+ pocket electrically doped (ED) tunnel FET” it is a theoretical concept or a device physically made (of some materials). If it is physically realized, some details about materials and dimensions must be provided.

In the sentence “To create a P+ source region, the PG1 terminal is biased at -0.7 V to increase the hole concentration to 1x1019 cm−3”, if the authors used the mentioned values (-0.7 V; 1x1019 cm−3) only as input parameters in a software program (for a theoretical FET concept/model), this must be clearly stated (the same mentions in the next paragraphs). Otherwise it can be understood that they built a device.

For the values used for the work functions of PG1 and PG2 (what kind of materials are referred to?), some references must be provided.

 2. Simulation Parameters and Approach

In Table 1, what is the meaning of “Conventional in-built N+ pocket ED-TFET” – it is a device or another model? However, a citation is needed for this “Conventional in-built N+ pocket ED-TFET”.

 3. Results and Discussions

At least one paragraph must be introduced to compare the results obtained by the authors for their “electrically doped (ED) PNPN tunnel FET” with a real device. Otherwise, all graphs remain just modeling with less interest for applicative work.

 4. Conclusions

The authors mention in the final sentence “As a result, the proposed device not 154 only achieves suppressed ambipolar currents, but also has a low thermal budget because of the electrically doped drain” the notion of “low thermal budget” but without numerical values. If they didn’t use the Software Package to simulate the “thermal budget” of their model FET this remains a presumption. So, either use the software to put some numerical data about the thermal behavior of their modeled FET or remove this notion.

 General remark:  the paper can be improved by comparing with the results (values) obtained on real FET-like devices.

Author Response

Dear editor,

   We are truly grateful to yours and other reviewers’ critical comments and thoughtful suggestions. Based on these comments and suggestions, we have made careful modifications on the original manuscript. We hope the new manuscript will meet your magazine’s standard. Below you will find our point-by-point responses to the reviewers’ comments/ questions:

Reviewer 2:

Abstract:

-          The concept of “electrical doped” from the sentence “In this paper, we propose and investigate an electrically doped (ED) PNPN tunnel FET…” must be clearly explained. Usually the term “doped” refers to an amount of another chemical element/compound added in order to obtain a certain effect.

  1. Thank you for your suggestion. The abstract has been modified based on your advice.

We present that proposed PNPN tunnel FETs can be realized without chemically doped junctions by utilizing polarity bias concept on a doped N+/P- starting structure.

-          All acronyms must be explained when firstly introduced, e.g. FET (field-effect transistor (FET))

  1. Thank you for your suggestion. We are so sorry for the mistake. The abstract has been modified based on your advice.

In this paper, we propose and investigate an electrically doped (ED) PNPN tunnel field effect transistor (FET), in which the drain side tunneling barrier width is effectively controlled to obtain suppressed ambipolar current.

Furthermore, 2-D Technology Computer Aided Design (TCAD) simulations have demonstrated that the introduction of the in-built N+ pocket yields a higher drive current and a steeper average SS when compared to conventional ED-TFET.

-          The concept/method of “calibrated TCAD simulations” must be explained as not usually used

  1. Thank you for your suggestion. We are so sorry for the mistake. The abstract has been modified based on your advice.

Using numerical device simulations, we demonstrate how the tunneling barrier width on the drain side can be influenced by several design parameters, such as the gap length between channel and drain (Lgap), the work-function of polarity gate, and dielectric material of the spacer.

  1. Introduction

The authors claim that “Based on polarity bias concept, we have recently reported an in-built N+ pocket electrically doped (ED) tunnel FET, which 31 provides a possible technique to overcome the difficulties in realizing a highly doped 32 narrow pocket [16].” In this sense, they are kindly asked to explain if “in-built N+ pocket electrically doped (ED) tunnel FET” it is a theoretical concept or a device physically made (of some materials). If it is physically realized, some details about materials and dimensions must be provided.

  1. The in-built N+ pocket electrically doped (ED) tunnel FET is a theoretical concept. The sentence has been modified based on your advice.

Based on polarity bias concept, we have recently reported a theoretical concept of an in-built N+ pocket electrically doped (ED) tunnel FET, which provides a possible technique to overcome the difficulties in realizing a highly doped narrow pocket [19].

In the sentence “To create a P+ source region, the PG1 terminal is biased at -0.7 V to increase the hole concentration to 1x1019 cm−3”, if the authors used the mentioned values (-0.7 V; 1x1019 cm−3) only as input parameters in a software program (for a theoretical FET concept/model), this must be clearly stated (the same mentions in the next paragraphs). Otherwise it can be understood that they built a device.

  1. Thank you for your suggestion. The manuscript has been modified based on your advice.

To create a P+ source region, the PG1 terminal is biased at -0.7 V to increase the hole concentration to 1◊1019 cm−3 in TCAD simulations.

For the values used for the work functions of PG1 and PG2 (what kind of materials are referred to?), some references must be provided.

  1. Thank you for your suggestion. The manuscript has been modified based on your advice.

Making the layer underneath the polarity gate intrinsic, the work functions of PG1 and PG2 in the proposed device with an ED drain are chosen to be 4.33 eV and 4.5 eV, which is referring to Ti and Mo material, respectively [24].

The work function of PG2 is set to 4.33eV, 4.5eV and 4.74eV, referring to Ti , Mo and Sb, respectively [24].

  1. 24. Haynes, M. Electron Work Function of the Elements. In CRC Handbook of Chemistry and Physics, 95th ed.; Lide, D. R., Bruno, T. J., Eds.; CRC Press: New York, NY, USA, 2008; pp. 12–1014.
  2. Simulation Parameters and Approach

In Table 1, what is the meaning of “Conventional in-built N+ pocket ED-TFET” – it is a device or another model? However, a citation is needed for this “Conventional in-built N+ pocket ED-TFET”.

  1. Thank you for your suggestion. The manuscript has been modified based on your advice.

Table 1. Comparison of device design.

Device Design

Conventional ED-TFET

Conventional in-built N+ pocket ED-TFET with uniform doped drain [19]

 Proposed In-built N+ pocket ED-TFET with an ED drain

Doping Profile of Beginning Structure (From Source to Drain)

N+- N+- N+

N+- P- - N+

N+- P-

Doping Profile of Final Stucture (From Source to Drain)

“P+”- N+- N+- N+

“P+”- N+- P- - N+

“P+”- N+- P- -“N+

Number of Polarity Gate

1

1

2

Location of Polarity Gate

Source Region

Source Region

Source and Drain Region

Table 2. Parameters used for device simulation.

Parameter

Conventional in-built N+ pocket ED-TFET [19]

In-built N+ pocket ED-TFET with an ED drain

Effective Gate Oxide Thickness (EOT)

0.8 nm

0.8 nm

Silicon Film Thickness (TSi)

10 nm

10 nm

Control Gate Length

50 nm

50 nm

Length of pocket (Lpocket)

5 nm

5 nm

Length of gap (Lgap)

-

5~25 nm

Source Doping

4×1019cm−3 (N+)

4×1019cm−3 (N+)

Channel Doping

1×1017cm−3 (P-)

1×1017cm−3 (P-)

Drain Doping

1×1017cm−3 (P-)

5×1018cm−3 (N+)

CG Work-function

4.74 eV

4.74 eV

PG1 Work-function

4.33 eV

4.33 eV

PG2 Work-function

-

4.5 eV

  1. Li, J.; Liu, Y.; Wei, S. F.; Shan, C. In-Built N+ Pocket Electrically Doped Tunnel FET With Improved DC and Analog/RF Performance. Micromachines, 2020, 11, 960. DOI: 10.3390/mi11110960.
  2. Results and Discussions

At least one paragraph must be introduced to compare the results obtained by the authors for their “electrically doped (ED) PNPN tunnel FET” with a real device. Otherwise, all graphs remain just modeling with less interest for applicative work.

  1. Thank you for your suggestion. To the best of our knowledge, electrically doped tunnel FET using the polarity bias concept has not yet been fabricated by any research groups. Polarity Controlled Silicon Nanowire Gate-All-Around FETs have been fabricated using the polarity bias concept [23]. Thus, we are unable to compare our simulation results with a real device using the same polarity bias technique. In addition, the tunnel FETs fabricated in references contain a variety of different structures, sizes and materials. However, in our opinion, the comparison between two devices with huge intrinsic structural differences is not convincing. The device structure we propose in this paper is just a theoretical concept by now. Therefore, all the results are obtained by simulation. However, based on Ref. [4], we have validated our simulation model using a nonlocal band-to-band tunneling (BTBT) model. These simulated results may be slightly higher or lower than those of the experimental results, but this does not have much impact on our findings, because the focus of this paper is not on the exact values of currents, but more on the general trends in the effect of design parameters (e.g. Lgap) on ambipolar current and relative results of ambipolar current in ED PNPN TFET with an ED drain and with a uniformly doped drain. As a result, this will not impact the conclusion of this paper.
  2. Boucart, K.; Ionescu, A. M. Double-gate tunnel FET with high-k gate dielectric. IEEE Trans. Electron Dev. 2007, 54, 1725–1733. DOI: 10.1109/TED.2007.899389.
  3. De Marchi, M.; Sacchetto, D.; Frache, S.; Zhang, J.; Gaillardon, P.-E.; Leblebici, Y.; De Micheli, G. Polarity control in double-gate, gate-all-around vertically stacked silicon nanowire FETs. in Proceedings of IEEE Electron Devices Meeting, San Francisco, CA, USA, 10 Dec. 2012. DOI: 10.1109/IEDM.2012.6479004.
  4. Conclusions

The authors mention in the final sentence “As a result, the proposed device not 154 only achieves suppressed ambipolar currents, but also has a low thermal budget because of the electrically doped drain” the notion of “low thermal budget” but without numerical values. If they didn’t use the Software Package to simulate the “thermal budget” of their model FET this remains a presumption. So, either use the software to put some numerical data about the thermal behavior of their modeled FET or remove this notion.

  1. Thank you for your suggestion. We are so sorry for the mistake. This notion has been removed based on your advice.

As a result, the ambipolar current of the proposed device can be effectively suppressed by choosing appropriate design parameters.

 General remark:  the paper can be improved by comparing with the results (values) obtained on real FET-like devices.

Chan Shan

Round 2

Reviewer 1 Report

The comments are addressed in the paper. 

Reviewer 2 Report

The paper can be now accepted for publication